# Hamstring harvest results in significantly reduced knee muscular protection during side-step cutting two years after anterior cruciate ligament reconstruction

Jason M. Konrath[1,2]*, Bryce A. Killen[1], David J. Saxby[1], Claudio Pizzolato[1], Ben A. Kennedy[3], Christopher J. Vertullo[1,4], Rod S. Barrett[1], David G. Lloyd[1]

1 School of Allied Health Sciences and Menzies Health Institute Queensland, Griffith University, Gold Coast, Queensland, Australia, 2 Principia Technology, Crawley, Western Australia, Australia, 3 QScan Radiology, Gold Coast, Queensland, Australia, 4 Knee Research Australia, Gold Coast, Queensland, Australia

* jason.konrath@principia.com.au

**Data Availability Statement:** All data is available from figshare database. DOI: 10.6084/m9.figshare. 22139915.

## Abstract

The purpose of this study was to determine the effect of donor muscle morphology following tendon harvest in anterior cruciate ligament (ACL) reconstruction on muscular support of the tibiofemoral joint during sidestep cutting. Magnetic resonance imaging (MRI) was used to measure peak cross-sectional area (CSA) and volume of the semitendinosus (ST) and gracilis (GR) muscles and tendons (bilaterally) in 18 individuals following ACL reconstruction. Participants performed sidestep cutting tasks in a biomechanics laboratory during which lower-limb electromyography, ground reaction loads, whole-body motions were recorded. An EMG driven neuro-musculoskeletal model was subsequently used to determine force from 34 musculotendinous units of the lower limb and the contribution of the ST and GR to muscular support of the tibiofemoral joint based on a normal muscle-tendon model (Standard model). Then, differences in peak CSA and volume between the ipsilateral/contralateral ST and GR were used to adjust their muscle-tendon parameters in the model followed by a recalibration to determine muscle force for 34 musculotendinous units (Adjusted model). The combined contribution of the donor muscles to muscular support about the medial and lateral compartments were reduced by 52% and 42%, respectively, in the adjusted compared to standard model. While the semimembranosus (SM) increased its contribution to muscular stabilisation about the medial and lateral compartment by 23% and 30%, respectively. This computer simulation study demonstrated the muscles harvested for ACL reconstruction reduced their support of the tibiofemoral joint during sidestep cutting, while the SM may have the potential to partially offset these reductions. This suggests donor muscle impairment could be a factor that contributes to ipsilateral re-injury rates to the ACL following return to sport.

**Funding:** Funding was given via an Australian Post graduate award (APA) scholarship. This has been added. The funders had no role in study design, data collection and analysis, decision to publish, or preparation of the manuscript.

**Competing interests:** The authors declare that no competing interests exist.

## Introduction

Anterior cruciate ligament (ACL) injuries are common in many sports involving sudden change of direction [1]. Worryingly, secondary rupture (i.e., re-rupture of the reconstructed ACL) has been found to occur at a higher frequency than primary ACL rupture, with 27% of patients suffering a second ACL rupture within ten years of their initial reconstruction [2, 3]. Surgical intervention is generally required to restore stability to the tibiofemoral joint following ACL rupture, with a graft being used to replace the original ACL and restore passive stability. Many orthopaedic surgeons favour the quadruple bundle hamstring auto-graft using the semitendinosus (ST) and gracilis (GR) tendons [4, 5]. However, previous studies have revealed the morphology of these donor muscles to be substantially altered following tendon harvest, including reduced cross sectional area, reduced volume, and retraction of the musculotendinous junction [5, 6]; with either no tendon, or abnormal tendon regeneration resulting in knee flexion and internal tibial rotation weakness [6]. Unsurprisingly, donor muscles are substantially weaker post-surgery, possibly due to their loss of physiological cross-sectional area, which may have implications for stability of the tibiofemoral joint during return to sport where sudden change of direction is common.

It has been previously established that ACL injuries typically occur during non-contact manoeuvres including sidestep cutting and single leg landing [7, 8]. Combined knee flexion, valgus, and internal rotation moments have been shown to place high strain on the ACL [9]. Laboratory studies have further demonstrated sidestep cutting produces higher valgus and internal rotation moments than straight running [10–12]. Given that muscles surrounding the tibiofemoral joint have the potential to support external moments applied to the knee, loss of donor muscle physiological cross sectional area due to tendon harvest may compromise the ability of these muscles to support of the tibiofemoral joint during sidestep cutting. Moreover, this could be a factor that contributes to the high re-injury rates of the ACL following return to sport from ACL reconstruction (ACLR). To our knowledge, no studies to date have investigated the mechanical effects of altered donor muscle morphology on muscular support for the knee during sporting movements. Such an investigation requires a computational modelling approach to determine individual muscle forces and their corresponding moments.

Individual muscle forces can be estimated by computational neuromusculoskeletal (NMS) models. Electromyography (EMG)-driven models use measured muscle activity in conjunction with an anatomical model to estimate muscle forces and implicitly account for a participants individual activation patterns [13]. Importantly, EMG-driven models have been shown to well predict external joint moments during gait [13–15] and the tibiofemoral joint contact forces from patients with instrumented knee implants [16, 17]. Thus, EMG-driven NMS models enable the exploration of the effects of ACLR on donor muscle contributions to tibiofemoral stability. To isolate the effects of altered muscle-tendon properties in donor muscles (i.e., ST and GR), comparisons must be performed under the same kinematics, underlying muscle activation patterns, and external loading conditions for pertain to each individual. As such, a comparison with controls is not appropriate, rather, *in silico* approaches are needed.

The purpose of this study was to determine the effect of simulating subject-specific donor muscle morphology changes on their contribution of muscular support to the tibiofemoral joint during sidestep cutting, given an identical set of kinematic and external loading conditions. We hypothesised the ST and GR would reduce their contribution to muscular support at the tibiofemoral joint during sidestep cutting following ACLR, indicating this could be a factor that contributes to ipsilateral re-injury rates to the ACL following return to sport. It was envisaged that the findings from this study could be used to establish better intervention strategies to prevent re-rupture of the ACL following ACLR with the use of ST and GR auto-grafts.

## Materials and methods

### Participants

A total of 18 patients (13 male, 5 female, mean age 28 ± 7 years, mean body mass 82 ± 16 kg) that underwent a quadrupled hamstring autograft using ST and GR tendons were recruited for the study. Inclusion criteria included (1) Isolated ACL injury sustained without any other concomitant knee ligament surgery; (2) ACL reconstruction with the use of a quadrupled autologous ST and GR graft at least two years previously, and no more than 4 years previously; (3) Between the age of 18 and 45; (4) The ability to comply with testing protocol. Exclusion criteria were: (1) contraindications to magnetic resonance imaging (MRI); (2) complex knee injuries with additional ligament surgery or meniscal injury; (3) any previous ACL or lower extremity surgery. Using data from Konrath et al. [6], the estimated effect size of post ACL reconstructed donor muscle morphology changes was large (Cohen's d > 0.8). The current investigation had an estimated 99% power to detect group differences in contributions to muscular support between standard and adjusted models with an alpha of 0.05. Surgeons followed the standardised protocol for a quadrupled hamstring graft in ACLR presented in detail in a previous publication [6]. The tendons were left attached to their distal insertion points, while an open end Linvatec (Linvatec, Largo FL) tendon stripper was used to release the tendons proximally from their muscular attachment points, using a cut technique rather than a push technique [18], to a length of 22 cm in females and 24 cm in males. Ethics approval was obtained through Human Research Ethics Committee of Griffith University (Ref No: PES/36/10/HREC) and all participants provided their written informed consent prior to data collection.

### Experimental protocol

Bilateral lower limb muscle morphology was assessed for each patient using MRI scans acquired from a nearby radiology clinic. All patients subsequently underwent an assessment of their sidestep cutting technique in a motion capture Biomechanics Laboratory within 1 week of their MRI scan.

### Muscle morphology

The methods used to acquire the muscle and tendon morphology have been described in detail in a previous publication [6] and will only be described briefly here. Participants were instructed to lie supine in a 3 Tesla MRI scanner (Philips Medical Systems). Axial T1-weighted 3D fast field echo sequences (FFE) spanning both legs were acquired from the level of the iliac crest to the ankle mortise, the FFE images provided excellent visibility of the muscle's outer margin for manual segmentation. Axial proton density 2D turbo spin echo sequences (PD) were collected from the level of the mid-thigh to below the tibial tuberosity spanning both legs, the PD images provided excellent visibility of the tendons for manual segmentation.

Image analysis was performed using Mimics software (Materialise, v17, Belgium). The ST and GR muscles of the surgical leg were first inspected for signs of a regenerated tendon. A tendon was classified as being a regenerated tendon if tendon tissue was visible below the musculotendinous junction, as well as having a clear insertion into the tibia. Tendons that were not visible below the musculotendinous junction, were classed as no regeneration. The volume and peak cross-sectional area (CSA) were determined for the ST and GR of the surgical and contralateral leg by manually tracing the contour of each respective muscle/tendon on each axial slice. These contours were then used to build a digitized three-dimensional mesh model of the respective muscle and tendon using the Mimics *Calculate3D* tool which calculates

volume. The Mimics *Fit Centreline* tool was used to create an inter-centroid line for each mesh model, with a plane orthogonal to the inter-centroid line used to calculate the CSA.

## Sidestep cutting

Participants were first familiarized with the sidestep cut task in accordance with previous publications [11, 12], until they felt comfortable with the movement and subsequently performed a minimum of four trials. Participants approached the force plate at a speed of 4–4.5 m/s and performed the sidestep cut off the surgical leg at an angle of 45˚ from the approach direction. It was ensured the stance foot was neither turned in, nor turned out and that they maintained an upright torso, facing in the direction of travel. To ensure participants cut at the desired angle, tape was placed along the ground and visual inspection ensured they changed direction accordingly. Speed was controlled by measuring the velocity of the left anterior-superior-iliac-spine optical marker in the direction of travel. A trial was considered successful provided the foot of the surgical leg landed wholly and central on the force platform, as well as achieving the desired speed.

Whole body three-dimensional motion during sidestep cutting was recorded at 100 Hz using a 12-camera VICON MX motion analysis system (Vicon, Oxford, UK). Reflective surface mounted markers were placed on prominent anatomical landmarks in accordance with the UWA marker set [19], with 3-marker clusters attached to the upper-limb and 10-marker clusters used on lower-limb segments to gain higher fidelity in assessing knee motion [20]. Static [21] and functional [19] tasks were performed to help identify hip and knee joint centres. Ground reaction forces (GRF) were sampled at 2000 Hz from two force plates (Advanced Mechanical Technology, Watertown, USA).

Activation patterns from 16 muscles on the surgical leg were recorded at 2000 Hz using a wireless EMG system (Zerowire, Wave Wireless EMG) with bipolar Ag/AgCl surface electrodes (Duo-Trode, Myotronics, USA). The recording were: medial hamstring group (semimembranosus (SM)/semitendinosus (ST)), biceps femoris (BF) which comprised biceps femoris long head (BFLH) and biceps femoris short head (BFSH), adductor group (AG), rectus femoris (RF), vastus lateralis (VL), vastus medialis (VM), gracilis (GR), tensor fascia latae (TFL), sartorious (SR), gluteus maximus (GMax), gluteus medius (GMed), medial gastrocnemius (MG), lateral gastrocnemius (LG), soleus (SL), tibialis anterior (TA) and peroneals (PR).

Marker trajectories, GRF, and EMG were processed using Matlab (The Mathworks, USA). All digital filtering was performed using $2^{nd}$ order Butterworth filters passed bidirectionally to remove phase effects. Marker trajectories and GRF were filtered using 10 Hz low-pass cut-off frequency. The EMG were band-pass filtered (30–500 Hz), full wave rectified, and then low pass filtered (6 Hz cut-off) to yield linear envelopes for each muscle [13], and subsequently scaled to their maximum value determined from maximum voluntary contractions under a range of contractile conditions.

## The standard NMS model

To isolate the effects of different donor muscle-tendon properties under the same kinematics, underlying muscle activation patterns, and external loading conditions, we used the surgical leg with unadjusted muscle parameters as the standard model [22]. This assumes the pre-surgical morphology of the donor muscles is not significantly different to the morphology from the contralateral muscles, which has been shown in previous literature [5].

A generic full-body musculoskeletal anatomic model [23, 24] was used in OpenSim v3.3 [25]. The markers identifying prominent anatomic landmarks recorded from the calibration trials, and the subsequent estimated joint centres, were used to scale and register the

generic OpenSim model to the participant's static pose. A standard NMS model was created for each participant to estimate musculotendon forces during the stance phase of the side-step cut assuming there was no morbidity to ST and GR using the calibrated EMG-informed NMS model (CEINMS) [14]. The CEINMS has been described in detail in a previous publication [14] and so is only described briefly. Each of the 34 MTU were modelled as a contractile element in series with a compliant tendon [26]. The tendon was modelled using a non-linear function normalised to tendon slack length ($l_t^s$) and the peak active muscle force [26]. The contractile element model consisted of generic force-length, force-velocity, and parallel elastic functions, in which final MTU force ($F_{MTU}$), is dependent on each MTU maximum isometric force ($F_m^{MAX}$), optimal fibre length ($l_m^o$), and pennation angle at optimal fibre length ($\emptyset_m^o$).

Calibration consisted of two steps: anatomical scaling followed by functional scaling [14, 27, 28]. For anatomical scaling, markers identifying the prominent anatomic landmarks were recorded from the calibration trials, and the subsequent estimated joint centres were used to scale and register the generic OpenSim model to the participant's static pose. The anatomical bodies, associated bone geometries and muscle lengths/paths were first linearly scaled using the estimated joint centres [28]. In addition to linear scaling, the muscle optimal fibre and tendon slack lengths for each MTU were optimised such the normalised muscle fibre and tendon operating curves were maintained, while preserving the overall length of the MTU across a range of physiological lower-limb joint angles [27].

Functional scaling was then performed within CEINMS by adjusting the EMG-driven model parameters such the least squared differences were minimised between model-predicted joint moments and the experimental joint moments from inverse dynamics analysis. Within CEINMS, a *closed loop* calibration was performed that used one trial of a range of gait tasks (walk, run, sidestep cut) to adjust the parameters such the CEINMS predicted joint moments tracked the experimentally measured joint moments about four degrees of freedom (DOF) [14] whilst ensuring the muscles stayed within their physiological range of fibre lengths. The four DOF used for calibration were hip adduction-abduction (HAA), hip flexion/extension (HFE), knee flexion/extension (KFE), and ankle dorsi/plantar flexion (AFE) [15]. Following calibration, CEINMS operated as an *open-loop predictive system* for each of the sidestep cut trials to calculate muscle forces and joint moments as a function of activation and 3D joint angles [14].

### The adjusted muscle model

An adjusted model was also created for each participant with modifications to the ST and GR Hill-type muscle-model parameters. Each adjusted model was identical to its standard model with the following exception: the ($l_m^o$), ($l_t^s$), and strength coefficients that were initially created for the ST and GR in the standard model were adjusted based on measured morphological differences between the surgical and contralateral limb as described by Konrath et al. [22]. Briefly, the volume ($V_m$) and peak cross sectional area ($CSA_m$) differences of the ST and GR between surgical and contralateral legs were used to calculate adjustments to the ($l_m^o$), ($l_t^s$), and strength coefficients of the ST and GR in the adjusted model [22].

The functional calibration was then repeated, however, the adjusted $l_m^o$, adjusted $l_t^s$, and adjusted strength coefficients for the ST and GR were fixed and not permitted to change during the optimisation. Following this functional calibration, new parameters were obtained for the 32 non-donor MTU within the model (34 minus the ST and GR which were not optimized) [14, 22]. The *open-loop* prediction system was then performed to calculate MTU forces and moments for each of the 34 MTU in the adjusted model [14].

### The adjusted tendon model

Out of the 18 participants tested, 7 demonstrated regeneration of both ST and GR tendons, 5 demonstrated regeneration of only one tendon (4 GR, 1 ST) and in the remaining 6 neither tendon showed any sign of regeneration. In the cases in which tendons did not regenerate, the ST and GR were found to scar down into the fascia of the SM. For the regenerated tendons, we assumed the normalised linear stiffness was the same as the standard model. The tendon was modelled as a non-linear function normalised to the adjusted tendon slack length ($l_t^s$) and the adjusted maximum isometric force [26]. The muscle tendon pathway for the donor muscles within the anatomical model was not changed.

For the ST and GR muscles that did not show signs of tendon regeneration, it was assumed the muscle did not exert force on the tendon or surrounding tissues. Strength coefficients for these muscles were set to zero and model calibration was performed to determine parameters for the 32 non-donor MTU with the model. The *open-loop* prediction system was then run to obtain muscle forces and moments for each of the 34 MTU.

### Statistical analysis

A repeated measures general linear model was used to assess the effect of model (standard versus adjusted) on joint moments, peak moments from individual knee muscles, and average muscular support of valgus knee moments about the medial and lateral compartments across stance during sidestep cutting. Further, point-to-point differences in joint and muscle moments and muscle contributions to tibiofemoral stability between the standard and adjusted model were determined for the medial knee muscles throughout stance phase, using a paired sample T-test at each point of the gait cycle [29]. As an additional analysis, a between group general linear model with a-priori contrasts were also performed to determine group differences in the knee flexion moment generated from muscles and semimembranosus contribution to muscular support between each tendon regeneration group (none versus both ST and GR tendon regeneration). All statistical analysis was performed using SPSS version 22 (SPSS Inc, Chicago, Ill). Significance was accepted for $p < 0.05$.

### Results

The standard and adjusted model produced nearly identical estimates of the net varus and valgus muscle moments about the medial and lateral compartments during sidestep cutting (Fig 1A and 1B). There were no significant differences in either the peak or average net muscle moments. About the medial compartment, the external and net muscle moments apply a valgus moment. In contrast, about the lateral compartment the external and net muscle moments

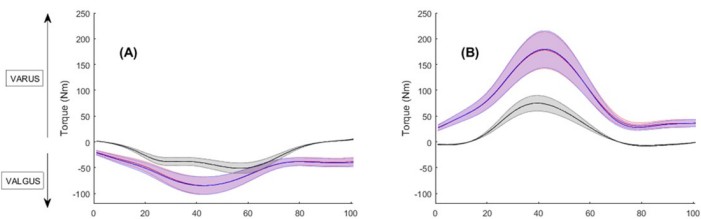

**Fig 1. The mean frontal plane external moments (black) and net muscle moments for the standard model (red) and adjusted model (blue) across stance phase about the (A) medial compartment and (B) lateral compartment.** Shaded regions indicate ± one standard error. Varus moments appear as positive, whilst valgus moments appear as negative.

apply a varus moment. External moments and net muscle moments were larger in magnitude about the lateral compartment than they were about the medial compartment (Fig 1A and 1B).

Compared to the standard model, peak moments generated about the medial compartment over stance were significantly smaller for the ST (p = 0.003), GR (p<0.0001), RF (p<0.0001) and BFSH (p = 0.0002) but larger for the SM (p<0.0001) and VL (p = 0.0001) in the adjusted model (Fig 2A). Compared to the standard model, peak moments generated about the lateral compartment during stance were significantly smaller for the ST (p = 0.0005), GR (p<0.0001), SR (p<0.0001) and RF (p<0.0001) but larger for the SM (p<0.0001) and VL (p = 0.0002) in the adjusted model (Fig 2B).

A point-by-point analysis of medial muscle moments about the medial and lateral condyles throughout stance phase is presented in supplementary Figs 3 and 4, respectively. For the medial compartment, the ST and GR showed significantly smaller moments in the standard compared to adjusted models throughout the entire stance phase, with the SR showing a smaller moment through the first 70% of stance (Fig 3). The SM showed significantly larger moments in the standard compared to adjusted models throughout the majority of stance phase (90%) (Fig 3). For the lateral compartment, the ST and GR showed significantly smaller moments in standard compared to adjusted models throughout the entire stance phase, with the SR showed a smaller moment through the first 70% of stance (Fig 4). The SM showed significantly larger moments in standard compared to adjusted models throughout the entire stance phase (Fig 4).

Compared to the standard model, contributions to muscular support about the medial compartment averaged over stance phase were significantly reduced for ST (p<0.0001), GR (p = 0.0003), RF (p<0.0001) and BFSH (p = 0.01) but increased for SM (p<0.0001) and VL (p<0.0001) in the adjusted model (Fig 5A). With respect to the lateral compartment, ST (p<0.0001) and GR (p<0.0001) contributions to muscular support averaged over stance phase for the lateral compartment were significantly reduced in the adjusted compared to standard model, while the contributions of SM (p = 0.0001) and VL (p<0.0001) significantly increased (Fig 5B).

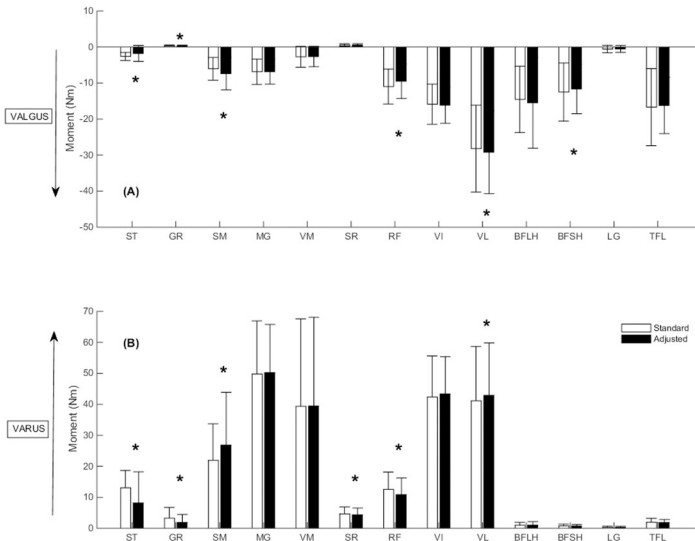

**Fig 2. The peak moment for the individual knee muscles across stance phase about the (A) medial compartment and (B) lateral compartment.** Error bars represent ± one standard deviation. (*) denotes statistical significance. Varus moments appear as positive, whilst valgus moments appear as negative.

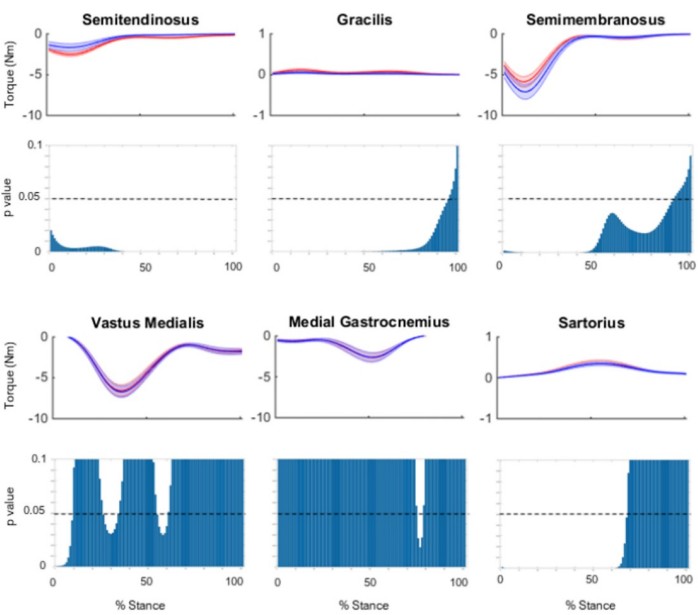

**Fig 3. The mean frontal plane moments about the medial compartment, for the standard model (red) and adjusted model (blue) across stance phase.** Shaded regions indicate ± one standard error. Point by point p-values are also shown for each point of stance phase.

Figs 6 and 7 show the mean contribution and standard error for each model to muscular support about the medial and lateral compartments, respectively, for the ST, GR and SM; plotted against the knee flexion angle across the stance phase of side-step cutting. A point-by-point analysis is also presented, showing differences between standard and adjusted model. In

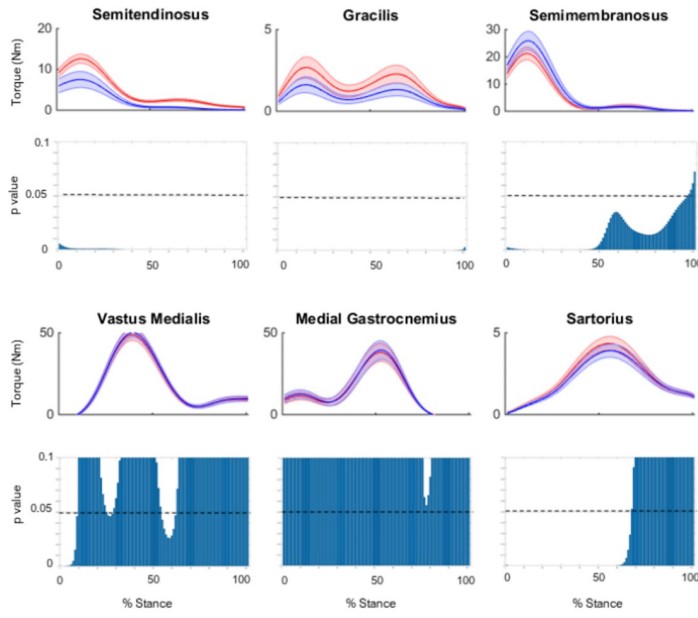

**Fig 4. The mean frontal plane moments about the lateral compartment, for the standard model (red) and adjusted model (blue) across stance phase.** Shaded regions indicate ± one standard error. Point by point p-values are also shown for each point of stance phase.

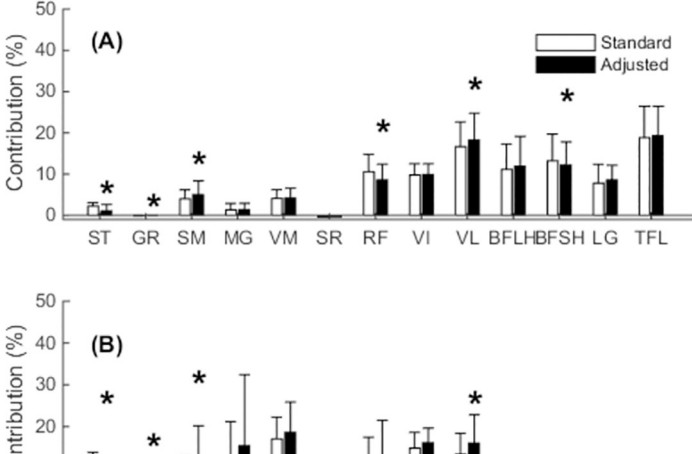

**Fig 5. The mean contribution of the individual knee muscles to muscular support averaged across stance phase about the (A) medial compartment and (B) lateral compartment.** Error bars represent ± one standard deviation. (*) denotes statistical significance.

both compartments, a reduction of support can be observed for the adjusted model of the ST and GR throughout stance, with larger deficits seen for the lateral compartment, while an increase in support can be observed for the SM in the first 50% of stance (Figs 6 and 7).

The no tendon regeneration group was found to have a significantly smaller knee flexion moment produced by the muscles averaged over stance when compared to both groups (no

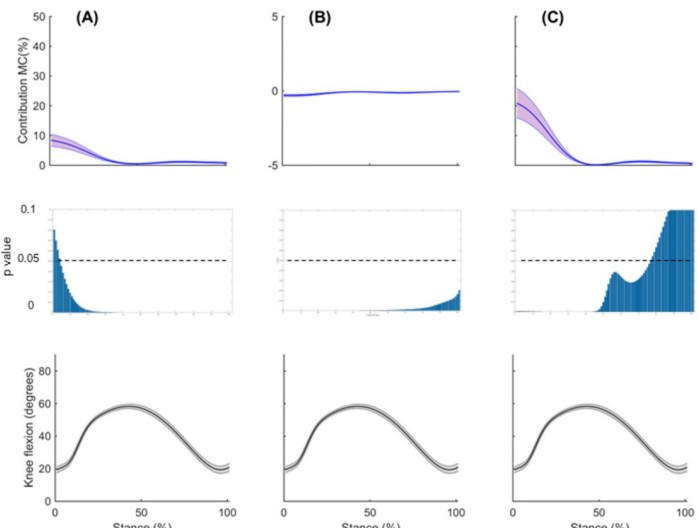

**Fig 6. The mean contribution to muscular support about the medial compartment for the standard model (red) and adjusted model (blue) across stance phase, with the knee flexion angle during stance phase for the (A) semitendinosus (B) gracilis and (C) semimembranosus.** Shaded regions indicate ± one standard error. Point by point p-values are also shown for each point of stance phase. Note the different scale of the vertical axis for the GR.

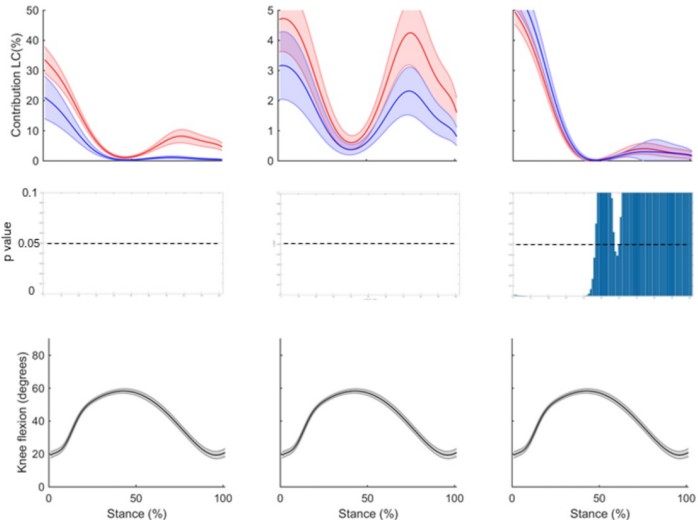

**Fig 7. The mean contribution to muscular support about the lateral compartment for the standard model (red) and adjusted model (blue) across stance phase, with the knee flexion angle during stance phase for the (A) semitendinosus (B) gracilis and (C) semimembranosus.** Shaded regions indicate ± one standard error. Point by point p-values are also shown for each point of stance phase. Note the different scale of the vertical axis for the GR.

tendons: 0.58 ± 0.22 Nm/kg, both tendons: 0.73 ± 0.18 Nm/kg) (p = 0.007). In addition, the SM showed larger contributions to muscular support about the lateral compartment in the no tendon regeneration group versus both tendon regeneration group (no tendon: 13 ± 9%, both tendon: 7 ± 5%) (p = 0.002). There were no differences in SM contributions to muscular support about the medial compartment between tendon regeneration groups (no tendon: 4.3 ± 3%, both tendon: 3.7 ± 3%).

## Discussion

This study investigated the effect of subject-specific changes to donor muscle morphology following ACL reconstruction on contributions to muscular support of the tibiofemoral joint. The combined contributions of ST and GR to muscular support about the medial and lateral compartments were reduced by 52% and 41.7%, respectively, in the models adjusted to account for the altered morphology of donor muscles in the surgical limb. In contrast, SM on the surgical limb increased its contribution to muscular support about the medial and lateral compartments by 22.5% and 30%, respectively. Overall, these findings confirm the contributions of ST and GR to muscular support of the tibiofemoral joint are markedly reduced during sidestep cutting, but may have the potential ability to be partially compensated by the SM to support the same external joint moments in the frontal plane.

The standard and adjusted models produced nearly identical estimates of the net varus and valgus muscle moments about the medial and lateral compartments during sidestep cutting, with larger external moments and net muscle moments about the lateral compared to the medial compartment. Winby et al. [30] demonstrated ample muscular support of the external moments about the medial and lateral compartments generated during walking, and stated a negative residual moment implied that other soft tissues were not required to preserve joint stability. In addition, Saxby et al. [31] showed knee muscles play an important role in supporting the valgus moments about the lateral compartment during sidestep cutting and may act to prevent medial compartment liftoff. In the case of a sidestep performed safely in laboratory

conditions, it is unlikely that the external loads would be high enough to cause injury, hence why we observe no statistical difference between the total muscular support. The reduced muscle forces and moments from the donor muscles can be compensated by muscles with equally appropriate anatomical moment arms to support the same external loading conditions. From a mechanical point of view, an ACL injury will occur when the loads applied to the ligament are greater than its ability to sustain the load [32]. In the case of many unanticipated sporting scenarios, the external loads can be much greater [10] and the reductions from the donor muscles may become more concerning.

The group without tendon regeneration had a 20% smaller knee flexion moment averaged across stance when compared to the group with both tendons regenerated. This supports the findings from previous studies investigating post ACL reconstruction knee strength and morphology, in which a 30% loss of knee flexion strength, as well as greater deficits in donor muscle volume and peak CSA in patients that do not experience tendon regeneration [6]. D'Alessandro et al. [18] reported a lower incidence of hamstring pain and muscle strains in patients receiving a "cut" technique of harvesting the hamstring tendon rather than forcefully stripping the tendon away from the muscle belly. Surgical techniques that facilitate tendon regeneration may therefore be beneficial for restoring hamstring strength.

The peak moments generated by ST and GR about the medial and lateral compartments were substantially reduced in the adjusted compared to standard model. Further, the donor muscles reduced their combined contribution to muscular support about the medial and lateral compartments by an average of 52% and 42% over stance, respectively. Previous studies investigating the muscle activation strategies at the knee during pre-planned sidestep cutting have shown selective activation of muscles with moment arms best able to counter the external load [33]. At smaller knee flexion angles, medial muscles including ST and GR if activated appropriately, have anatomical moment arms capable of supporting external valgus moments [34]. This selective activation is believed to directly support the external moment and prevent condylar lift-off [33, 35]. Furthermore, larger contributions to muscular support by the donor muscles were observed during early stance (Fig 6). The hamstrings have greater potential to constrain anterior tibial translation when the knee is flexed [36], but decreased capacity to support varus and valgus loads in flexed compared extended positions [35]. This is due to large varus and valgus moment arms of the ST and GR at small knee flexion angles, the moment arms become smaller as the knee is flexed because the muscles become more parallel to the tibial plateau [34]. Collectively, this suggests the donor muscles may reduce their muscular support during early stance, when the knee is in more extended postures.

The non-donor muscles medial to the knee joint centre tended to produce the largest varus moments and muscular contribution to support about the lateral compartment, whilst the non-donor knee muscles lateral to the knee joint centre tended to produce the largest valgus moments and contribution to muscular support about the medial compartment. This finding supports the view that a muscle's contribution to muscular support of a valgus moment about the lateral compartment is synonymous with its ability to load the medial compartment of the knee joint [30]. The main contributors to muscular support about the lateral compartment were the quadriceps, medial hamstrings, and the MG, while the main contributors to muscular support about the medial compartment were the quadriceps, lateral hamstrings, LG, and the TFL. The MG may be an important contributor to muscular support about the lateral compartment. Selective activation of medial knee muscles including the MG have been reported and are believed to counter the external valgus load applied during sidestep cutting [33], while elevated gastrocnemius forces have also been reported to compensate for decreased hamstring forces during a single leg jump landing [37].

The SM was found to increase its contribution to muscular support about the medial and lateral compartment by 23% and 30%, respectively, between the adjusted and standard model. The increased support from the SM appears to occur in early stance phase when the knee is in more extended postures. Additionally, the no tendon regeneration group showed 85% larger contributions from the SM towards muscular support about the lateral compartment of the TFJ compared to group with both tendons regenerated. It has been suggested that hamstring activation before initial contact in landing tasks in an attempt to constrain anterior tibial translation [38]. The varus and valgus moment arms of the SM are also reduced as the knee flexes due to the muscles being more parallel with the tibial plateau [34]. This suggests the SM can compensate for reduced donor muscle function and may provide an explanation to why larger SM volumes have been reported in the surgical leg of ACLR patients when compared to the contralateral leg [6]. This may also explain why we saw some differences in the contributions of the RF, VL, and BFSH in the adjusted model. The SM has a larger flexion moment arm than the ST and GR, if its muscle forces are increased to improve support along the frontal plane, this may create a slight misbalance with respect to moments along the sagittal plane, which must be balanced via other quadricep and hamstring muscles. The human body has several kinematic DOF, from which it has more muscles than DOF, which can create several different combinations of muscle forces. The present study has demonstrated when we adjust the muscle forces of a muscle due to morbidity, its change in moment can have impacts on other DOF.

The muscular support of external valgus moments may play an important role in preventing secondary ACL rupture. The findings of the present study suggest the contributions of donor muscles to muscular support of valgus moments about the lateral compartment to be large during early stance while knee flexion angles were smaller. Previous studies have identified external valgus moments during sidestep cutting [11, 12], while cadaveric work has shown the ACL to be more susceptible to strain under combined flexion and valgus moments with small knee flexion angles [9]. Furthermore, video analysis of sports-related ACL injuries suggests the knee collapses into valgus and/or internal tibial rotation with knee flexion angles of less than 30˚ shortly after initial foot strike [7]. Given the donor muscles have the mechanical potential to contribute to support of external valgus moments at smaller knee flexion angles, their reduced function may infer higher loads to the ACL during early stance if they are not compensated by other non-donor knee muscles.

The present study was not without limitations. The anatomical models used in this study did not take into account subject-specific bone geometries or muscle pathways, although scaling factors were applied to the models based on each participant's anthropometry [27, 28]. For regenerated tendons, we assumed the donor MTU to follow the same path and insertion point as the contralateral MTU; however, the regenerated tendons generally reattach themselves more proximally which may reduce their varus and valgus moment arms. Future work should look at how the muscle paths and moment arms change for the donor MTU.

We further assumed the donor muscles to be independent actuators which were able to transmit force only through a tendon and not through surrounding musculature. Therefore, for donor muscles that did not show signs of a regenerated tendon, the MTU was removed from the post-surgery model. Fascial attachments to surrounding muscles, such as scarring down to the semimembranosus does occur and it is possible for this to allow transmission of force through the scarred down attachment to remaining muscles [39]. To our knowledge, no literature to date has tried to model this phenomenon using an NMS modelling approach. Further research would be needed to determine how this could be incorporated into NMS models and was beyond the scope of this study. However, despite this, we were able to show increased

contributions from the SM in the no-tendon regeneration group. If the ST and GR can scar down into the SM, this in fact would contribute additional force to the SM, which was observed in the model. There are other muscles with appropriate anatomical moment arms such as medial quadriceps and gastrocnemius that could increase contribution, however this wasn't observed. This gave us good confidence in the models.

## Conclusion

Changes in musculotendon properties of the donor muscles following ACL reconstruction using hamstring grafts were shown to reduce the contribution of donor muscles to muscular support of the tibiofemoral joint during sidestep cutting. To balance the same external valgus moments in our simulations, the deficits in donor muscle contributions to muscular support of the tibiofemoral joint were shown to have the ability to be partially offset by the SM. Increased contributions of the SM in the no tendon regeneration compared to regeneration group further confirmed that the SM can compensate for reduced donor muscle function. Results indicate donor site morbidity has a marked effect on the function of knee muscula-ture and its contribution to stability of the tibiofemoral joint. This may be considered a factor behind re-injury of the ACL for individuals that have undergone autologous hamstring grafts. Furthermore, due to its close proximity to the morbid donor muscles and its capacity to counter valgus moments at the knee in extended postures, targeting the SM muscle in training programs could be a beneficial strategy for compensating the loss of donor muscle function.

## Author Contributions

**Conceptualization:** Jason M. Konrath, Bryce A. Killen, David J. Saxby, Claudio Pizzolato, Ben A. Kennedy, Christopher J. Vertullo, Rod S. Barrett, David G. Lloyd.

**Data curation:** Jason M. Konrath, Bryce A. Killen, David J. Saxby, Claudio Pizzolato, Ben A. Kennedy, Christopher J. Vertullo.

**Formal analysis:** Jason M. Konrath, Bryce A. Killen, David J. Saxby, Claudio Pizzolato, Ben A. Kennedy.

**Funding acquisition:** David G. Lloyd.

**Investigation:** Jason M. Konrath, Bryce A. Killen, David J. Saxby, Claudio Pizzolato, Ben A. Kennedy.

**Methodology:** Jason M. Konrath, Bryce A. Killen, David J. Saxby, Claudio Pizzolato, Ben A. Kennedy.

**Project administration:** Jason M. Konrath, David J. Saxby, Christopher J. Vertullo, Rod S. Barrett, David G. Lloyd.

**Resources:** Christopher J. Vertullo, Rod S. Barrett, David G. Lloyd.

**Software:** Bryce A. Killen, Claudio Pizzolato, Ben A. Kennedy.

**Supervision:** David J. Saxby, Christopher J. Vertullo, Rod S. Barrett, David G. Lloyd.

**Writing – original draft:** Jason M. Konrath.

**Writing – review & editing:** Jason M. Konrath, Bryce A. Killen, David J. Saxby, Claudio Pizzo-lato, Ben A. Kennedy, Christopher J. Vertullo, Rod S. Barrett, David G. Lloyd.

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
