## [Decision Letter · Decision Letter 0]

6 Apr 2023

PONE-D-23-05205Hamstring harvest results in significantly reduced knee muscular protection during side-step cutting two years after anterior cruciate ligament reconstructionPLOS ONE

Dear Dr. Konrath,

Thank you for submitting your manuscript to PLOS ONE. After careful consideration, we feel that it has merit but does not fully meet PLOS ONE’s publication criteria as it currently stands. Therefore, we invite you to submit a revised version of the manuscript that addresses the points raised during the review process.

We look forward to receiving your revised manuscript.

Kind regards,

Yaodong Gu

Academic Editor

PLOS ONE

Journal Requirements:

'No. The funders had no role in study design, data collection and analysis, decision to publish, or preparation of the manuscript."

"No. The authors have decided that no competing interests exist."

5. We note that Figures 1 and 2 in your submission contain [map/satellite] images which may be copyrighted. All PLOS content is published under the Creative Commons Attribution License (CC BY 4.0), which means that the manuscript, images, and Supporting Information files will be freely available online, and any third party is permitted to access, download, copy, distribute, and use these materials in any way, even commercially, with proper attribution. For these reasons, we cannot publish previously copyrighted maps or satellite images created using proprietary data, such as Google software (Google Maps, Street View, and Earth). For more information, see our copyright guidelines: http://journals.plos.org/plosone/s/licenses-and-copyright.

a. You may seek permission from the original copyright holder of Figures 1 and 2 to publish the content specifically under the CC BY 4.0 license.  

Additional Editor Comments:

Please clarify the main novelty of this study.

Reviewers' comments:

Reviewer's Responses to Questions

**Comments to the Author**

1. Is the manuscript technically sound, and do the data support the conclusions?

Reviewer #1: Yes

Reviewer #2: Yes

2. Has the statistical analysis been performed appropriately and rigorously? 

Reviewer #1: Yes

Reviewer #2: No

3. Have the authors made all data underlying the findings in their manuscript fully available?

Reviewer #1: Yes

Reviewer #2: No

4. Is the manuscript presented in an intelligible fashion and written in standard English?

Reviewer #1: Yes

Reviewer #2: Yes

5. Review Comments to the Author

Reviewer #1: 1、“However, in order to segregate the effects of different donor muscle-tendon properties, comparisons must be performed under the same external loading conditions, kinematics and underlying muscle activation patterns that pertain to each individual.” Why is it important to distinguish the characteristics of different donor tendons here, as different tendons are used in the graft to restore the ACL, so should the description of the tendons here be modified to different donor muscles.

2、“(3) previous ACL or lower extremity surgery.” Please be more detailed here, e.g. ACLs over 4 years old.

3、“The ST and GR muscles of the surgical leg were first inspected for signs of a regenerated tendon.” Here it is determined whether it is a regenerated tendon or a part of the tendon that survived the original extraction of only part of the tendon.

4、Is there a reason why it is at this speed of 4-4.5 m/s?

5、“A standard NMS model was created for each participant to compute estimates of musculotendon forces during the stance phase of the sidestep cut.” Each subject has their own model after scaling, and how to achieve all have a standard NMS model.

Reviewer #2: Review comment

This manuscript entitled “Hamstring harvest results in significantly reduced knee muscular protection during side-step cutting two years after anterior cruciate ligament reconstruction” primarily aimed to determine the effect of subject-specific donor muscle morphology changes on their contribution toward muscular support of the tibiofemoral joint during sidestep cutting, given the same kinematic and external loading conditions. The authors bring an interesting study, but there are still some problems that cannot up this study to a publishing level. Some suggestions are listed in the specific comments below.

Specific comments:

1. Please modify and improve the quality of the keywords, as this will assist others when they are searching for information on your research topic. Avoid using “Hamstring” since it appears in the title.

2. Line 30-31, “The model was then rerun with subject-specific changes to muscle-tendon parameters for the donor muscles”, Please revise the methodological description in the Abstract as it is unclear.

3. Line 36-39, I suggest that the authors further emphasize the possible value of this study in their conclusion.

4. Line 42-43, the study from 2009 (ref. 1) does not support the conclusion that “have increased in incidence over the past decade”.

5. Line 50-51, “However, previous studies have revealed the morphology of these donor muscles to be substantially altered following tendon harvest…”, What specific changes in the morphology of the donor muscles occur following tendon harvest? Please clarify.

6. I could see that the authors provide some of the necessary research backgrounds to analyze the rationale of the present study in the introduction section, however, I did not see the necessity and potential significance of this study properly. What can this study add to this topic? The authors did not provide sufficient information about the research gap to further clarify the practical implications of this study.

7. Line 143, How was speed controlled or measured?

8. Line 248, More details need to be provided in the statistical analysis section. What specific parameters were analyzed? I suggest the authors perform a stronger approach and analysis as SPM analysis in order to observe the full time series of the biomechanical parameters among the assessments.

9. In the results section, please provide specific p-values rather than “p<0.05”.

10. Line 329-330, “… and are compensated for by the SM in order to support the same external joint moments in the frontal plane.”, More evidence needs to be provided to support the authors’ conclusion.

11. Line 442, The conclusions should be further strengthened based on the main findings of this study.

12. In summary, please ensure that your manuscript is prepared correctly (without any grammatical and spelling mistakes) and formatted before submitting a revision.

6. PLOS authors have the option to publish the peer review history of their article (what does this mean?). If published, this will include your full peer review and any attached files.

Reviewer #1: No

Reviewer #2: No

---

## [Author Response · Author response to Decision Letter 0]

25 Sep 2023

RESPONSE TO COMMENTS BY EDITOR 1 (E1)

E1 comment 1: Please clarify the main novelty of this study.

Author’s response 1: The present study was focused on whether or not donor muscle loss was a factor behind ipsilateral ACL re-injury. While reconstruction of the ACL can restore functional stability to the majority of patients, re-rupture of the reconstructed ACL has been found to occur at a higher rate than initial ACL injuries; with up to 26% of patients suffering an ACL re-injury within ten years of the initial reconstruction (Pinczewski et al., 2007). It is well known that ACL injuries typically occur during non-contact manoeuvers including sidestep cutting (Cochrane et al., 2007; Krosshaug et al., 2007). Combined flexion, valgus and internal rotation moments in particular have been shown to place high strain on the ACL (Markolf et al. 1995). Laboratory studies have further demonstrated that sidestep cutting produces higher valgus and internal rotation moments than straight running (Besier et al., 2001; Dempsey et al., 2007). While muscles surrounding the tibiofemoral joint have the potential to support external moments applied to the knee. It could be expected that loss of donor muscle physiological cross sectional area as a result of tendon harvest could compromise the ability of these muscles to contribute to muscular support of the tibiofemoral joint during sidestep cutting. Therefore, this could be a factor that contributes to ipsilateral re-injury rates to the ACL following return to sport from ACL reconstruction (ACLR).

The above points have been mentioned in the introduction, we have also added the following text into the introduction, to highlight how it may pertain to re-injury of the ACL following return to sport from ACLR, as well as needing a computational modelling approach to determine individual muscle forces and moments. We have also further strengthened the conclusion based on the main findings of the study

Line 62: 

Moreover, this could be a factor that contributes to the high re-injury rates of the ACL following return to sport from ACL reconstruction (ACLR). To our knowledge, no studies to date have investigated the mechanical effects of altered donor muscle morphology on muscular support for the knee during sporting movements. Such an investigation requires a computational modelling approach to determine individual muscle forces and their corresponding moments.

Line 83:

We hypothesised that the ST and GR would reduce their contribution to muscular support at the tibiofemoral joint during sidestep cutting following ACLR, indicating this could be a factor that contributes to ipsilateral re-injury rates to the ACL following return to sport. It was envisaged that the findings from this study could be used to establish better intervention strategies to prevent re-rupture of the ACL following ACLR with the use of ST and GR auto-grafts.

Line 482:

Results indicate donor site morbidity has a marked effect on the function of knee musculature and its contribution to stability of the tibiofemoral joint. This may be considered a factor behind re-injury of the ACL for individuals that have undergone autologous hamstring grafts. Furthermore, due to its close proximity to the morbid donor muscles and its capacity to counter valgus moments at the knee in extended postures, targeting the SM muscle in training programs could be a beneficial strategy for compensating the loss of donor muscle function.

 

RESPONSE TO COMMENTS BY REVIEWER 1 (R1)

R1 comment 1: “However, in order to segregate the effects of different donor muscle-tendon properties, comparisons must be performed under the same external loading conditions, kinematics and underlying muscle activation patterns that pertain to each individual.” Why is it important to distinguish the characteristics of different donor tendons here, as different tendons are used in the graft to restore the ACL, so should the description of the tendons here be modified to different donor muscles.

Author’s response 1: The present study focused on individuals that have had a quadrupled hamstring tendon autograft created from the semitendinosus (ST) and gracilis (GR) tendons. Other grafts can include the patellar tendon and quadriceps tendon, however there are a number of concerns with these grafts including those related to donor site morbidity (Pinczewski et al., 2007). As a result, hamstring tendon autografts have recently become a more common choice among orthopaedic surgeons. However, they are known to significantly alter the muscle-tendon properties of the ST and GR, with the deficits more pronounced in muscles with no tendon regeneration (Konrath et al., 2016). Therefore, the present study chose to focus on ST and GR grafts, as well as the effects of tendon regeneration vs no tendon regeneration, in these respective donor muscles.

The following sentence in the introduction (line 75) has been amended to highlight the focus on the altered muscle-tendon properties of the ST and GR.

To isolate the effects of altered muscle-tendon properties in donor muscles (i.e., ST and GR), comparisons must be performed under the same kinematics, underlying muscle activation patterns, and external loading conditions for pertain to each individual.

R1 comment 2: “(3) previous ACL or lower extremity surgery.” Please be more detailed here, e.g. ACLs over 4 years old.

Author’s response 2: As this was an exclusion criteria, any participant that had any previous ACL or lower extremity surgery would be excluded, regardless of time out from surgery etc.

The text has been amended to include any, to make this more clear

(3) Any previous ACL or lower extremity surgery.

R1 comment 3: “The ST and GR muscles of the surgical leg were first inspected for signs of a regenerated tendon.” Here it is determined whether it is a regenerated tendon or a part of the tendon that survived the original extraction of only part of the tendon.

Author’s response 3: Tendon regeneration was defined as having occurred if the tendon was visible below the musculotendinous junction, as well as having a clear path to it’s insertion onto the tibia. Tendons that were not visible below the musculotendinous junction, were classed as no regeneration.

The following text has been added to Line 128

A tendon was classified as being a regenerated tendon if tendon tissue was visible below the musculotendinous junction, as well as having a clear insertion into the tibia. Tendons that were not visible below the musculotendinous junction, were classed as no regeneration.

R1 comment 4: Is there a reason why it is at this speed of 4-4.5 m/s?

Author’s response 4: This is in accordance with sidestepping protocols in previous publications (Dempsey et al., 2007; Besier et al 2001)

The following text has been amended on line 140

Participants were first familiarized with the sidestep cut task in accordance with previous publications (10,12), until they felt comfortable with the movement and subsequently performed a minimum of four trials.

R1 comment 5: “A standard NMS model was created for each participant to compute estimates of musculotendon forces during the stance phase of the sidestep cut.” Each subject has their own model after scaling, and how to achieve all have a standard NMS model.

Author’s response 5: Calibration consisted of both anatomical and functional scaling. For anatomical scaling, markers identifying the prominent anatomic landmarks that were recorded from the calibration trials, and the subsequent estimated joint centres were used to scale and register the generic OpenSim model to the participant’s static pose. The anatomic bodies, associated bone geometries and muscle lengths/paths were first linearly scaled using the estimated joint centres (Winby et al., 2008). In addition to linear scaling, the muscle optimal fibre and tendon slack lengths for each MTU were optimised such that the normalised muscle fibre and tendon operating curves were maintained while preserving the overall length of the MTU across a range of physiological lower-limb joint angles (Modenese et al., 2015).

The following text has been added to line 195 to further elaborate on the anatomical scaling

Calibration consisted of two steps: anatomical scaling followed by functional scaling(14,25,26). For anatomical scaling, markers identifying the prominent anatomic landmarks that were recorded from the calibration trials, and the subsequent estimated joint centres were used to scale and register the generic OpenSim model to the participant’s static pose. The anatomic bodies, associated bone geometries and muscle lengths/paths were first linearly scaled using the estimated joint centres (26). In addition to linear scaling, the muscle optimal fibre and tendon slack lengths for each MTU were optimised such that the normalised muscle fibre and tendon operating curves were maintained while preserving the overall length of the MTU across a range of physiological lower-limb joint angles (25).

Functional scaling has been described on line 204.

RESPONSE TO COMMENTS BY REVIEWER 2 (R2)

R2 comment 1: Please modify and improve the quality of the keywords, as this will assist others when they are searching for information on your research topic. Avoid using “Hamstring” since it appears in the title.

Author’s response 1: The key terms have been changed to neuromusculoskeletal model, ACL injury, tendon harvest, graft, tibiofemoral joint support

R2 comment 2: Line 23-29, “The model was then rerun with subject-specific changes to muscle-tendon parameters for the donor muscles”, Please revise the methodological description in the Abstract as it is unclear.

The following text in the abstract has been amended

An EMG driven neuro-musculoskeletal model was subsequently used to determine force from 34 musculotendinous units of the lower limb and the contribution of the ST and GR to muscular support of the tibiofemoral joint based on a normal muscle-tendon model (Standard model).Then, differences in peak CSA and volume between the ipsilateral/contralateral ST and GR were used to adjust their muscle-tendon parameters in the model followed by a recalibration to determine muscle force for 34 musculotendinous units (Adjusted model). 

R2 comment 3: Line 36-39, I suggest that the authors further emphasize the possible value of this study in their conclusion.

Author’s response 3: We have added the following sentence in the abstract.

This suggests donor muscle impairment could be a factor that contributes to ipsilateral re-injury rates to the ACL following return to sport. 

R2 comment 4: Line 42-43, the study from 2009 (ref. 1) does not support the conclusion that “have increased in incidence over the past decade”.

Author’s response 4: We have removed the following sentence, as per the reviewer’s suggestion.

R2 comment 5: Line 50-51, “However, previous studies have revealed the morphology of these donor muscles to be substantially altered following tendon harvest…”, What specific changes in the morphology of the donor muscles occur following tendon harvest? Please clarify.

Author’s response 5: The following sentence has been amended as per the reviewer’s suggestion

However, previous studies have revealed the morphology of these donor muscles to be substantially altered following tendon harvest, including reduced cross sectional area, reduced volume and retraction of the musculotendinous junction (5,6); with either no tendon, or abnormal tendon regeneration resulting in knee flexion and internal tibial rotation weakness(6).

R2 comment 6: I could see that the authors provide some of the necessary research backgrounds to analyze the rationale of the present study in the introduction section, however, I did not see the necessity and potential significance of this study properly. What can this study add to this topic? The authors did not provide sufficient information about the research gap to further clarify the practical implications of this study.

Author’s response 6: We know from previous literature the morphological changes from hamstring harvest affect the knee flexion and internal rotation strength (Konrath et al., 2016), as well as donor muscle contributions to knee joint contact forces (Konrath et al., 2017). Given that medial hamstring muscles including the semimembranosus, semitendinosus and gracilis play an important role in the support of valgus moments at the knee, a known contributor to ACL loading and potential injury during sidestep cutting. Therefore, this could be a factor that contributes to ipsilateral re-injury rates to the ACL following return to sport from ACL reconstruction (ACLR).

This warrants investigation of how much the loss of donor muscle size and their corresponding force/moment production, influences the support of these valgus moments. Such knowledge may enable targeted training and rehabilitation programs to take place to try and improve the support of valgus moments during sidestep cutting. 

The following text has been added to the introduction, to highlight how it may pertain to re-injury of the ACL following return to sport from ACLR, as well as needing a computational modelling approach to determine individual muscle forces and moments.

Line 62: 

Moreover, this could be a factor that contributes to the high re-injury rates of the ACL following return to sport from ACL reconstruction (ACLR). To our knowledge, no studies to date have investigated the mechanical effects of donor muscle loss on muscular support for the knee during sporting movements. Such an investigation requires an in-silico modelling approach to determine individual muscle forces and their corresponding moments.

Line 83:

We hypothesised the ST and GR would reduce their contribution to muscular support at the tibiofemoral joint during sidestep cutting following ACLR, indicating this could be a factor that contributes to ipsilateral re-injury rates to the ACL following return to sport. It was envisaged that the findings from this study could be used to establish better intervention strategies to prevent re-rupture of the ACL following ACLR with the use of ST and GR auto-grafts.

R2 comment 7: Line 143, How was speed controlled or measured?

Author’s response 7: The following text has been added to the methods, as per the reviewer’s suggestion.

Line 147:

Speed was controlled by measuring the velocity of the left anterior-superior-iliac-spine optical marker in the direction of travel. A trial was considered successful provided the foot of the surgical leg landed wholly and central on the force platform, as well as achieving the desired speed.

R2 comment 8: Line 248, More details need to be provided in the statistical analysis section. What specific parameters were analyzed? I suggest the authors perform a stronger approach and analysis as SPM analysis in order to observe the full time series of the biomechanical parameters among the assessments.

Author’s response 8: We have performed additional statistical analyses as per the reviewer’s suggestion to observe the full time series of the biomechanical parameters, We have updated 2 figures to present this, and have also added 2 supplementary figures.

Figure 6, has now been updated and split into two separate figures (Figure 6 and 7 respectively). These figures now show point to point differences between standard and adjusted models, throughout the entire stance phase of the gait cycle. A paired sample T-test was used at each point of stance phase, with statistical significance being accepted for p<0.05, in accordance with methods described by Schwartz et al. (2004). Figures 6 and 7 look at the contribution of the respective muscles toward support of valgus moments about the medial and lateral compartment respectively.

Two additional figures have also been added (figures 3 and 4 respectively). These figures show the moments across stance phase for the medial muscles crossing the knee joint, that have the potential to provide support against valgus moments. With these moments, point to point differences between the standard and adjusted models were measured throughout the entire stance phase, with the methods described above

The following text has been added into the statistical analysis section (Line 250)

Further, point-to-point differences of moments and contributions between the standard and adjusted model were determined for the medial knee muscles throughout stance phase, using a paired sample T-test at each point of the gait cycle (28). 

The following text has been added to the results

Line 287:

A point-by-point analysis of medial muscle moments about the medial and lateral condyles throughout stance phase is presented in supplementary Figures 3 and 4, respectively. For the medial compartment, the ST and GR showed significantly smaller moments in the standard compared to adjusted models throughout the entire stance phase, with the SR showing a smaller moment through the first 70% of stance (Fig 3). The SM showed significantly larger moments in the standard compared to adjusted models throughout the majority of stance phase (90%) (Fig 3). For the lateral compartment, the ST and GR showed significantly smaller moments in standard compared to adjusted models throughout the entire stance phase, with the SR showed a smaller moment through the first 70% of stance (Fig 4). The SM showed significantly larger moments in standard compared to adjusted models throughout the entire stance phase (Fig 4).

Line 322:

Figures 6 and 7 show the mean contribution and standard error for each model to muscular support about the medial and lateral compartments, respectively, for the ST, GR and SM; plotted against the knee flexion angle across the stance phase of side-step cutting. A point-by-point analysis is also presented, showing differences between standard and adjusted model. In both compartments, a reduction of support can be observed for the adjusted model of the ST and GR throughout stance, with larger deficits seen for the lateral compartment, while an increase in support can be observed for the SM in the first 50% of stance (Fig 6 and Fig 7).

R2 comment 9: In the results section, please provide specific p-values rather than “p<0.05”.

Author’s response 9: The specific p values have been added in the manuscript, as per the reviewer’s suggestion. For p values that were less than 10^(-4), p < 0.0001 was added.

R2 comment 10: Line 329-330, “… and are compensated for by the SM in order to support the same external joint moments in the frontal plane.”, More evidence needs to be provided to support the authors’ conclusion.

Author’s response 10: We have changed the wording to now say the semimembranosus (SM) has the potential ability to be partially compensated by the SM in order to support the same external joint moments in the frontal plane.

We believe sufficient evidence exists to suggest this, in particular with the new graphs and point-by-point comparisons suggested by R2 comment 8. Increased support from the SM appears to occur in early stance phase, at a similar stage to when the decreased support is observed from the donor muscles, when the knee is in more extended postures (Figures 6 and 7). This is likely due to the moment arm of the SM in the varus/valgus plane being larger as the knee is extended, due to the muscles being more perpendicular with the tibial plateau (Lloyd and Buchanan, 2001). Furthermore, these modelling results are also consistent with results observed in Konrath et al. (2016), which showed the SM of the surgical leg to have larger muscle volume than that of the contralateral leg. 

The following text has been amended (Line 359):

Overall these findings confirm that the contributions of ST and GR to muscular support of the tibiofemoral joint are markedly reduced during sidestep cutting, and have the potential ability to be partially compensated by the SM in order to support the same external joint moments in the frontal plane.

The above points have also been mentioned in the discussion from line 435 to line 444, as well as the new figures being added as per suggestions from R2 comment 8.

R2 comment 11: Line 442, The conclusions should be further strengthened based on the main findings of this study.

Author’s response 11: The following text has been added to the conclusion to further strengthen the main findings of the present study (Line 483).

Changes in musculotendon properties of the donor muscles following ACL reconstruction using hamstring grafts were shown to reduce the contribution of donor muscles to muscular support of the tibiofemoral joint during sidestep cutting. To balance the same external valgus moments in our simulations, the deficits in donor muscle contributions to muscular support of the tibiofemoral joint were shown to have the ability to be partially offset by the SM. Increased contributions of the SM in the no tendon regeneration compared to regeneration group further confirmed that the SM can compensate for reduced donor muscle function. Results indicate donor site morbidity has a marked effect on the function of knee musculature and its contribution to stability of the tibiofemoral joint. This may be considered a factor behind re-injury of the ACL for individuals that have undergone autologous hamstring grafts. Furthermore, due to its close proximity to the morbid donor muscles and its capacity to counter valgus moments at the knee in extended postures, targeting the SM muscle in training programs could be a beneficial strategy for compensating the loss of donor muscle function. 

R2 comment 12: In summary, please ensure that your manuscript is prepared correctly (without any grammatical and spelling mistakes) and formatted before submitting a revision.

Author’s response 12: Changes made, as suggested.

---

## [Decision Letter · Decision Letter 1]

2 Oct 2023

Hamstring harvest results in significantly reduced knee muscular protection during side-step cutting two years after anterior cruciate ligament reconstruction

PONE-D-23-05205R1

Dear Dr. Konrath,

We’re pleased to inform you that your manuscript has been judged scientifically suitable for publication and will be formally accepted for publication once it meets all outstanding technical requirements.

Kind regards,

Yaodong Gu

Academic Editor

PLOS ONE

Additional Editor Comments (optional):

Well done!

Reviewers' comments:

Reviewer's Responses to Questions

**Comments to the Author**

1. If the authors have adequately addressed your comments raised in a previous round of review and you feel that this manuscript is now acceptable for publication, you may indicate that here to bypass the “Comments to the Author” section, enter your conflict of interest statement in the “Confidential to Editor” section, and submit your "Accept" recommendation.

Reviewer #1: All comments have been addressed

Reviewer #2: All comments have been addressed

2. Is the manuscript technically sound, and do the data support the conclusions?

Reviewer #1: Yes

Reviewer #2: Yes

3. Has the statistical analysis been performed appropriately and rigorously? 

Reviewer #1: Yes

Reviewer #2: Yes

4. Have the authors made all data underlying the findings in their manuscript fully available?

Reviewer #1: Yes

Reviewer #2: Yes

5. Is the manuscript presented in an intelligible fashion and written in standard English?

Reviewer #1: Yes

Reviewer #2: Yes

6. Review Comments to the Author

Reviewer #1: After careful revision by the authors, this manuscript has been greatly improved and all issues have been addressed. And I believe that this manuscript can be successfully published.

Reviewer #2: (No Response)

7. PLOS authors have the option to publish the peer review history of their article (what does this mean?). If published, this will include your full peer review and any attached files.

Reviewer #1: No

Reviewer #2: No

---

## [Editor Report · Acceptance letter]

4 Oct 2023

PONE-D-23-05205R1 

Hamstring harvest results in significantly reduced knee muscular protection during side-step cutting two years after anterior cruciate ligament reconstruction 

Dear Dr. Konrath:

I'm pleased to inform you that your manuscript has been deemed suitable for publication in PLOS ONE. Congratulations! Your manuscript is now with our production department. 

Kind regards, 

on behalf of

Professor Yaodong Gu 

Academic Editor

PLOS ONE